# Exploring Variability of Free Asparagine Content in the Grain of Bread Wheat (*Triticum aestivum* L.) Varieties Cultivated in Italy to Reduce Acrylamide-Forming Potential

**DOI:** 10.3390/plants12061349

**Published:** 2023-03-16

**Authors:** Andrea Tafuri, Melania Zuccaro, Stefano Ravaglia, Raul Pirona, Stefania Masci, Francesco Sestili, Domenico Lafiandra, Aldo Ceriotti, Elena Baldoni

**Affiliations:** 1National Research Council (CNR), Institute of Agricultural Biology and Biotechnology (IBBA), via E. Bassini 15, 20133 Milano, Italy; andrea.tafuri@ibba.cnr.it (A.T.); melania.zuccaro@ibba.cnr.it (M.Z.); raul.pirona@cnr.it (R.P.); aldo.ceriotti@ibba.cnr.it (A.C.); 2SIS Società Italiana Sementi, Via Mirandola 5, 40068 San Lazzaro di Savena, Italy; s.ravaglia@sisonweb.com; 3Department of Agriculture and Forest Sciences, University of Tuscia, Via San Camillo de Lellis, 01100 Viterbo, Italy; masci@unitus.it (S.M.); francescosestili@unitus.it (F.S.); lafiandr@unitus.it (D.L.)

**Keywords:** bread wheat, free asparagine, acrylamide-forming potential, food safety

## Abstract

Acrylamide, a suspected human carcinogen, is generated during food processing at high temperatures in the Maillard reaction, which involves reducing sugars and free asparagine. In wheat derivatives, free asparagine represents a key factor in acrylamide formation. Free asparagine levels in the grain of different wheat genotypes has been investigated in recent studies, but little is known about elite varieties that are cultivated in Italy. Here, we analysed the accumulation of free asparagine in a total of 54 bread wheat cultivars that are relevant for the Italian market. Six field trials in three Italian locations over two years were considered. Wholemeal flours obtained from harvested seeds were analysed using an enzymatic method. Free asparagine content ranged from 0.99 to 2.82 mmol/kg dry matter in the first year, and from 0.55 to 2.84 mmol/kg dry matter in the second year. Considering the 18 genotypes that were present in all the field trials, we evaluated possible environment and genetic influences for this trait. Some cultivars seemed to be highly affected by environment, whereas others showed a relative stability in free asparagine content across years and locations. Finally, we identified two varieties showing the highest free asparagine levels in our analysis, representing potential useful materials for genotype x environment interaction studies. Two other varieties, which were characterized by low amounts of free asparagine in the considered samples, may be useful for the food industry and for future breeding programs aimed to reduce acrylamide-forming potential in bread wheat.

## 1. Introduction

Wheat (*Triticum* spp. L.) is one of the world’s most important crops, contributing an estimated 18.6% to global daily calorie intake and 19.8% to global daily protein intake in 2018 (http://www.fao.org/faostat/en/ (accessed on 7 January 2023)). The world currently produces around 770 mmt of wheat annually [1]. A key reason for the importance of wheat as a staple crop is its ability to be used as the main ingredient in a plethora of products, such as breads, noodles, couscous, and pasta. In Southern Europe, about 5 million hectares were used for wheat cultivation in 2021, with a production of about 22 million tonnes. In 2020, in the same geographic area, per capita consumption of wheat was estimated at about 115 kg (http://www.fao.org/faostat/en/ (accessed on 7 January 2023)). In Italy, wheat represents an important part of the national cereal market with more than 7 million tonnes produced in 2021, as shown also by a 3.3% increase in production compared to the average of the last 5 years (https://agridata.ec.europa.eu/extensions/DashboardCereals/CerealsProduction.html (accessed on 6 December 2022)). 

Wheat breeding is principally aimed to elevate yield potential with stability, reduce the requirements for water, fertilizers, and other inputs without affecting productivity, increase the ability of wheat to grow on marginal lands, and protect wheat production from emerging threats of climate change [2]. Today, one of the objectives of wheat breeding is the improvement of its nutritional quality and health value [3,4]. A major issue related to food production that can be faced through specific breeding programs is the reduction of acrylamide, which is generated in a wide range of foods, including wheat derivatives [5,6]. Acrylamide (C_3_H_5_NO) is a white, odourless, water-soluble molecule and is rapidly absorbed through the skin and from the gastro-intestinal tract. Once metabolised, it produces glycidamide (C_3_H_5_NO_2_), which forms adducts with DNA both in vitro and in vivo in human and animal tissues [7]. Acrylamide is classed in group 2a as a probable carcinogenic to humans by the International Agency for Research on Cancer (IARC) [8]. Acrylamide is generated by Maillard cascade reactions during high-temperature processing of foods (frying, baking, roasting, toasting) [9,10,11,12]. These reactions involve free reducing sugars, such as glucose, fructose, and maltose, and free asparagine (fAsn), which gives the carbon skeleton. The fAsn concentration has been identified as the main determinant of acrylamide-forming potential in wheat and other cereals [12,13,14]. For this reason, a major recommendation in the food industry to reduce acrylamide levels cereal-based products is the use of grain with low fAsn content [15]. Free amino acids generally account for about 5% or less of the total nitrogen content of wheat grain, with fAsn accounting for 10% or less of the total [16]. Consequently, this amino acid was of little scientific interest until the demonstration that it is a precursor of acrylamide during cooking. Only in recent years has an increasing number of studies aimed to compare the level of fAsn in the seeds of different wheat genotypes and to identify the factors influencing this trait. Variability in fAsn accumulation in the grain is high among wheat genotypes [15,17,18,19]. It is well demonstrated that this trait is highly influenced by environmental factors, such as abiotic and biotic stresses [16], sulphur or nitrogen availability [11,13,20,21,22,23,24], and poor disease control [21,25]. Nevertheless, the accumulation of fAsn in wheat seed is a specific trait showing good heritability [26,27]. In addition, genotype versus environmental (GxE) effects influence fAsn content in wheat seeds [13,15,18,28,29]. Thus, both breeding and genetics may play an important role in selecting and developing cultivars accumulating low and stable fAsn levels across different growing environments, and then to mitigate the acrylamide risk in wheat products.

Most of the studies about fAsn in wheat grain have been related to bread wheat (*Triticum aestivum* L.) genotypes that are principally cultivated in the U.S., the UK, Canada, and Australia [15,18]. These studies underlined that many differences in fAsn concentration were due to the growing environment and crop management, although some cultivars may be considered as low asparagine accumulating genotypes for the involvement of genetic factors. Contrastingly, there is little information on fAsn concentration in bread wheat varieties cultivated in Italy. To our knowledge, only Corol and collaborators [18] included in their study two bread wheat varieties cultivated and commercialized in Italy (i.e., Blasco and Palesio), both classified as low fAsn genotypes. Considering the importance of bread wheat in the Italian agricultural system and food industry, the aim of this study was to perform a preliminary evaluation of fAsn content in the grain of selected elite varieties of bread wheat, relevant for the Italian market, to set the base of a characterization of this genetic material in terms of acrylamide-forming potential. We considered a total of 54 genotypes, that are widely cultivated in Italy, and three growing locations over two years to evaluate possible influences of both environmental and genetic factors. Hence, our analysis allowed for the identification of cultivars showing relatively low and stable fAsn levels. These genotypes may be used in future breeding programs aimed to mitigate the acrylamide risk in wheat derivatives.

## 2. Results and Discussion

### 2.1. Free Asparagine Quantification in Six Field Trials Over Two Years

Field trials of selected bread wheat elite varieties relevant for the Italian market were performed over two years (2018–2019 and 2019–2020) at three locations across Southern, Central, and Northern Italy (Foggia, Grosseto, Voghera, respectively). There were 54 varieties altogether over the two trial seasons: 36 genotypes in 2018–2019 and 36 genotypes in 2019–2020, with 18 varieties being present in both. The genotypes are listed in Table 1, grouped according to the Italian bread wheat classification based on chemical and rheological properties [30,31]. Bread wheat is unique for the production of leavened food, bread and biscuits in particular. The classification of different genotypes relies on several physical, chemical, and rheological characteristics. Worldwide, various classification systems are principally based on objective evaluations of gluten proteins. In Italy, bread wheat is classified into five categories, depending on its end use [31]. The specific classes are: (i) FF = improver wheat (“Frumento di Forza”in Italian), varieties suitable for manufacturing highly leavened products with a very resistant gluten network; (ii) FPS = superior bread making wheat (“Frumento Panificabile Superiore”), varieties with consistent milling and baking performance; (iii) FP = ordinary bread making wheat (“Frumento Panificabile”), varieties with bread-making potential but not suited to all grists; (iv) FB = wheat for biscuits (“Frumento da Biscotto”), soft varieties used for biscuits, breakfast cereals, cakes, and similar products. The variety list for 2018–2019 comprised six FF varieties, five FPS varieties, twenty FP varieties, and five FB varieties (Table 1). The variety list for 2019–2020 comprised six FF varieties, nine FPS varieties, eighteen FP varieties, and three FB varieties (Table 1).

A recent study underlined that there are no significant differences in free asparagine levels among different classes related to rheological characteristics [15]. Nevertheless, it may be useful to indicate the classification of the analysed genotypes to help farmers and the food industry in decision making. 

Wholemeal flour was produced from the grain harvested from the six field trials. The flour was then analysed for fAsn concentration; the results are reported in Appendix A. The fAsn content in the considered samples ranged from 0.99 to 2.82 mmol/kg dry matter (d.m.) for 2018/2019 and from 0.55 to 2.84 mmol/kg d.m. for 2019/2020. 

Considering the first year of field trials (2018–2019, Figure 1), Giambologna, classified as FF, showed the lowest amount of fAsn, reaching 0.99 mmol/kg d.m. in the Foggia field trial, whereas RGT_Montecarlo, classified as FPS, showed the highest amount of fAsn reaching 2.82 mmol/kg d.m. (Appendix A) in the Voghera field trial.

Some cultivars displayed fAsn amounts lower than 1.5 mmol/kg in the three field trials, i.e., Giambologna (classified as FF), Blasco (classified as FPS), Ascott, LG_Absalon, Sothys_CS, SY_Cicerone (all classified as FP), and the two FB genotypes Oregrain and Santorin. On the other hand, the two FPS genotypes Lancillotto and RGT_Montecarlo, as well as the FB variety Cosmic, showed fAsn amounts higher than 2 mmol/kg in the three field trials. 

In the second year of field trials (2019–2020, Figure 2), KWS_Coli, a FP variety, showed the lowest amount of fAsn, reaching 0.55 mmol/kg d.m. in the Grosseto field trial, whereas the FPS cultivar SY_Starlord showed the highest amount of fAsn, reaching 2.84 mmol/kg d.m. (Appendix A) in the Voghera field trial. 

Some cultivars displayed fAsn amounts lower than 1.5 mmol/kg in the three field trials: the three FF varieties Aiace_VST, Bologna, and Giorgione, the FPS genotype Solindo_CS, and six FP varieties (i.e., Antigua, KWS_Coli, KWS_Lazuli, LG_Arnova, Poker_VST, and SY_Cicerone). No bread wheat variety showed fAsn amounts higher than 2 mmol/kg in the three field trials of the second year. Compared to the first year of field trials, the data of the second year seemed to display a general lower amount of fAsn. In particular, the Grosseto field trial presented very low levels of fAsn, with most of the bread wheat varieties showing a range of fAsn amounts between 0.5 to 1 mmol/kg. 

Overall, our results were comparable with data obtained in previous works [13,32,33], where fAsn levels ranged between 0.68 and 4.77 mmol/kg d.m. when plants were grown in optimal conditions and treated against pathogens infections. As expected, a variability among the considered genotypes was observed. Although the reported data were preliminary and needed to be confirmed by further evaluations, our analysis suggested that both the environment and genetics may affect fAsn accumulation in the grain of the 54 considered varieties. As reported above, 18 genotypes were sown in both growing seasons during 2018/2019 and 2019/2020. We then focused on the data related to these 18 genotypes to evaluate the possible effect played by the environment and/or by the genotype on fAsn accumulation in the considered bread wheat panel.

### 2.2. Evaluation of the Influence of Growing Environments on Free Asparagine Accumulation

To evaluate a possible influence of the environment, the measurements of fAsn content were compared among the six field trials, considering the data related to each genotype as a whole (Figure 3).

In the two field trials in Foggia, the median values of fAsn levels in the 18 genotypes were very similar: 1.48 and 1.47 mmol/kg d.m. for the first and the second year, respectively. Similarly, grain harvested in Voghera showed similar median values of fAsn levels in the 18 genotypes between the two field trials: 1.75 and 1.55 mmol/kg d.m. for the first and the second year, respectively. Overall, the Foggia and the Voghera samples showed comparable levels of fAsn. Differently, the genotypes grown in Grosseto in the second year accumulated a lower amount fAsn (median value 0.95 mmol/kg d.m.) compared to those grown in the previous year in the same location (median value 1.81 mmol/kg d.m.). This data is consistent with the previous observation related to the 36 genotypes of the second year (Figure 2), where the Grosseto field trial presented very low levels of fAsn. Interestingly, the samples derived from the field trial in Grosseto in 2018–2019 and 2019–2020 showed the highest and the lowest levels of fAsn content, respectively, compared to those measured in the six field trials considered here. The large effect of crop management and other environmental factors, both alone and interacting with genetic factors (G × E), has a reasonable impact in fAsn accumulation [12]. It has been well demonstrated how differences in growth location and/or growth year may influence fAsn accumulation due to environmental conditions [19,34,35]. Many authors have shown how differences in sulphur or nitrogen availability strongly affect fAsn accumulation [11,13,15,20,21,22,23,24]. In our case, the considered field trials were conducted in the frame of an Italian network of experimental fields (i.e., the Italian “National Network of Common Wheat Variety Comparison”; in Italian “Rete Nazionale di Confronto Varietale del Frumento Tenero”). This network shared similar agricultural practices among field trials, including protocols for fertilization. For this reason, nitrogen and sulphur supply resulted in similar results among the field trials (Appendix A) and probably did not influence the low fAsn level observed in the 2019–2020 Grosseto field trial. The observed differences in fAsn accumulation in this field trial, when compared to the previous year and to the other locations of Foggia and Voghera, were probably due to environmental factors, such as rainfall and soil water availability, or specific climatic conditions, which highly affected the fAsn accumulation in wheat seeds. This data is in agreement with a previous analysis, which reported that the level of precipitation and temperature during grain development (between heading and harvest) had the greatest effect on fAsn concentration in mature grain [18].

### 2.3. Evaluation of the Influence of Genetic Factors on Free Asparagine Accumulation

To preliminary characterize a possible effect played by the genotype in fAsn accumulation of the selected bread wheat panel, we compared the fAsn content of the 18 common genotypes considering all of the six measurements (three locations for two years; Figure 4).

The box plots in Figure 4 show a high variability in fAsn content among years and locations for some genotypes. In particular, RGT_Montecarlo and Lancillotto showed the highest variability by accumulating fAsn with differences between maximum and minimum values of about 4.06 and 3.64 times, respectively. Other genotypes appeared more stable in terms of fAsn content across year and locations. LG_Arnova and LG_Ascona showed the lowest variability in the amount of fAsn, with differences between maximum and minimum values of about 1.44 and 1.63 times, respectively.

Considering the observed levels of fAsn in each genotype in the analysed field trials, SY_Cicerone showed a relatively low and stable amount of fAsn, ranging from 0.63 to 1.42 mmol/kg d.m. (median value 1.21 mmol/kg d.m.). Similarly, the genotype Sothys_CS showed a fAsn content ranging from 1.02 to 1.98 mmol/kg d.m. (median value 1.26 mmol/kg d.m.). If these data could be confirmed in further analyses, these varieties may be utilised to mitigate the acrylamide risk in wheat products and to develop low fAsn accumulator genotypes in future breeding programs. On the other hand, Stromboli showed relatively high fAsn levels, ranging between 1.39 and 2.45 mmol/kg d.m. (median value 2.19 mmol/kg d.m.). RGT_Montecarlo fAsn values were highly variable between different trials, but reached the highest amount of fAsn (2.82 mmol/kg d.m.) of the entire panel and showed a median value (2.07 mmol/kg d.m.) comparable to Stromboli.

The variability in fAsn accumulation among genotypes has been well reported in the literature [13,15,17,20,27,33,35]. Despite the documented effect of environmental factors, the accumulation of fAsn in wheat seeds is a specific trait showing good heritability [15,26,27]. To date, crop management strategies, including ensuring sufficient sulphur in the soil during cultivation [23] and avoiding pathogen infection [21,25], are the most common strategies to reduce fAsn concentrations. Nevertheless, a further crucial point to control fAsn content in wheat seeds is the identification and utilization of wheat genotypes with a strong genetic component in this trait [5]. A recent study highlighted that a natural deletion of a key gene encoding the asparagine synthetase of class 2 of the B genome (TaASN-B2) can reduce the amount of fAsn in wheat grain [36]. The presence and absence of this gene was investigated in a panel of common wheat varieties, showing that free asparagine concentrations in field-produced grain were, on average, lower in varieties lacking TaASN-B2 [36]. In this regard, some breeding programs are already ongoing to reduce the amount of fAsn in wholemeal and refined flour by modulating the expression of key genes, such as ASN genes, that are involved in its synthesis inside the kernel.

Previous studies were able to identify specific bread wheat genotypes that were considered low fAsn accumulators and confirmed that selection for low asparagine content is possible [19,21,27,28]. Hence, the analysis of fAsn accumulation in several wheat genotypes sown in different locations may be crucial to identify stable genotypes, as well as specific genetic traits associated with fAsn accumulation. 

## 3. Materials and Methods

### 3.1. Plant Material and Field Trials

This study was conducted on 54 bread wheat varieties (*Triticum aestivum* L.) that were included in the Italian “National Network of Common Wheat Variety Comparison” (In Italian: “Rete Nazionale di Confronto Varietale del Frumento Tenero”) coordinated by the Council for Agricultural Research and Economics (CREA). The field trials were conducted in three Italian locations (Foggia, Grosseto, and Voghera) for two crop years (2018/2019 and 2019/2020). Geographical coordinates and sowing and harvesting dates are reported in Table 2 for the six field trials.

The applied fertilization inputs are reported in Appendix A. Weather data were collected throughout the field trials for each location (Appendix A).

### 3.2. Flour Deproteination

Whole-grain samples were ground through a 0.5 mm grid diameter mesh using a laboratory mill (Cyclotec CT293, FOSS, Hilleroed, Denmark) to obtain a homogeneous powder. Wholemeal flour (5 g) was added to 15 mL of 1 M perchloric acid and shaken for 30 min at room temperature. The suspension was centrifuged at 5300 RCF for 3 min at 4 °C to produce a clear extract. Clear supernatant was brought to pH 8 by adding 2 M KOH and adjusted to 50 mL in a graduated cylinder. The samples were refrigerated in an iced bath for 30 min. Samples were then centrifuged at 5300 RCF for 30 min at 4 °C. Clear supernatant was filtered through a 0.45 μm syringe filter and 2 mL was transferred into a 2 mL tube and stored at −20 °C for subsequent analysis. 

### 3.3. Free Asparagine Quantification

The quantification of fAsn levels in deproteinated wholemeal flour samples was conducted according to Lecart and co-workers [32] with slight modifications, using a K-ASNAM L-Asparagine/L-Glutamine/Ammonia kit (Megazyme, Wicklow, Ireland).

This kit foresees the use of three enzymatic reactions.
L-glutamine + H_2_O → L-glutamate + NH^+^_4_(1)
NH^+^_4_ + 2-oxoglutarate + NADPH → L-glutamate + NADP^+^ + H_2_O(2)
L-asparagine + H_2_O → L-aspartate + NH^+^_4_(3)

Practically, 200 µL of the deproteinated sample was collected and mixed with 100 µL of the pH 4.9 buffer (offered in the K-ASNAM kit) and 10 µL of glutaminase (reaction 1). After a 5 min time lapse at room temperature, 150 µL of the second buffer at pH 8.0 was added together with 100 µL of NADPH and 600 µL of distilled water. After a delay of 10 min at room temperature, 10 µL of glutamate dehydrogenase was added (reaction 2). Subsequently, both the ammonium ions of the sample and those generated by the first reaction react with 2-oxoglutarate and NADPH. The absorbance at 340 nm was then measured to measure the conversion of NADPH to NADP+. After measurement, 10 µL of asparaginase was added (reaction 3). In this step, asparagine is hydrolysed to aspartate and ammonium ions by the action of asparaginase. This reaction liberates ammonium ions, which enter the second reaction and lead to an additional fall in absorbance at 340 nm, which is stoichiometric with the concentration of asparagine in the sample. The absorbance was recorded using a Jasco V-630 spectrophotometer (JASCO, Tokyo, Japan). The concentration of fAsn in the sample was calculated as reported by the manufacturer.

To check the repeatability of the quantification protocol, the same deproteinated whole-grain sample of wheat (cv Palesio) was routinely added to each quantification procedure and measured as the control. All the fAsn measurements of this control sample are reported in Appendix A. The variability among these measurements was estimated at about 0.41 mmol/kg and considered as the intrinsic error of the quantification method. The average value of Palesio was 1.48 ± 0.10 mmol/kg.

## 4. Conclusions

To our knowledge, this work reports the first comprehensive analysis related to fAsn accumulation in seeds of bread wheat elite varieties relevant for the Italian market. Although these data need to be validated in further field trials, this analysis represents a starting point for the characterization of this genetic material. The food industry requires varieties that can be relied upon to produce grain with low fAsn amounts over a range of environments and harvest years [15,37]. Despite the environmental role in fAsn accumulation, we identified specific bread wheat varieties that showed good stability among years and locations in terms of fAsn concentration. Our data also suggested that the varieties SY_Cicerone and Sothys_CS, with low amounts of fAsn in the grain and a low influence from the environment, may be useful for the food industry to mitigate the acrylamide risk in wheat derivatives. In contrast, Stromboli and RGT_Montecarlo, which showed high fAsn levels in our analysis, may be useful for studying the GxE interaction in this trait. In conclusion, these materials can be useful for further studies to dissect the mechanisms involved in fAsn accumulation in wheat seeds, as well as for studies aimed to reduce the acrylamide-forming potential of wheat varieties. 

## Figures and Tables

**Figure 1 plants-12-01349-f001:**
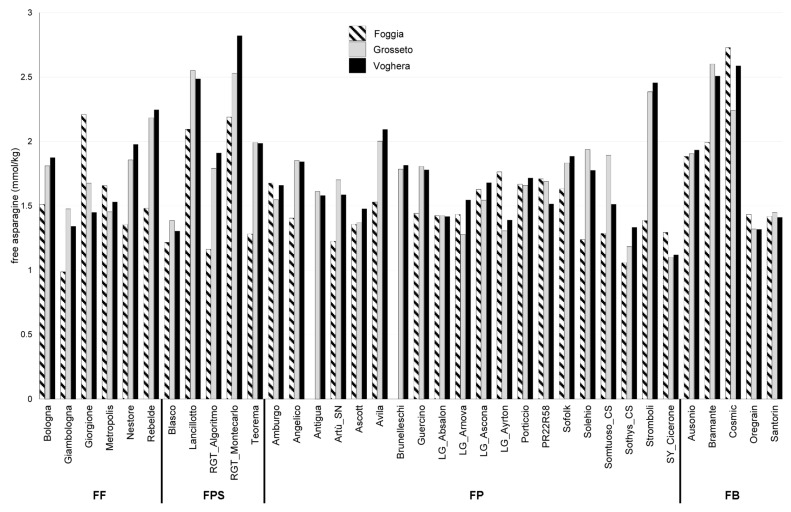
Concentration of fAsn in the grain of 36 bread wheat varieties grown in 2018–2019 in the three locations Foggia, Grosseto, and Voghera. FF: improver wheat; FPS: superior bread making wheat; FP: ordinary bread making wheat; FB: wheat for biscuits.

**Figure 2 plants-12-01349-f002:**
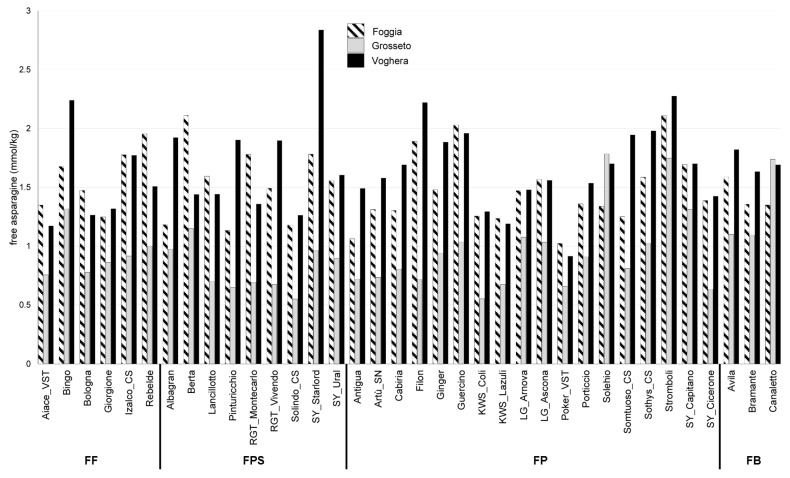
Concentration of fAsn in the grain of 36 bread wheat varieties grown in 2019–2020 in the three locations Foggia, Grosseto, and Voghera. FF: improver wheat; FPS: superior bread making wheat; FP: ordinary bread making wheat; FB: wheat for biscuits.

**Figure 3 plants-12-01349-f003:**
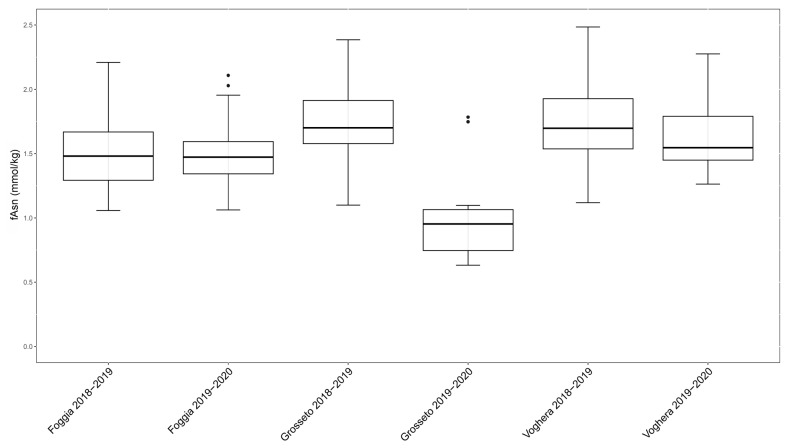
Box plots of fAsn measurements in the six considered field trials, considering the data from the 18 genotypes common to the six field trials.

**Figure 4 plants-12-01349-f004:**
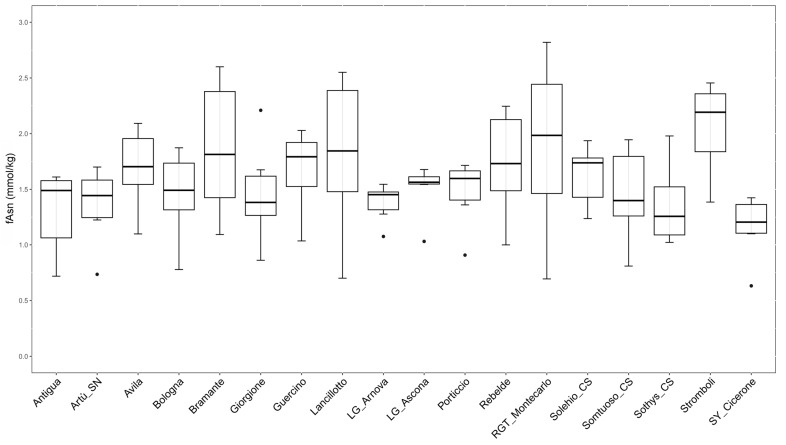
Box plots of fAsn measurements from the 18 genotypes present in the six field trials.

**Table 1 plants-12-01349-t001:** List of the 54 analysed varieties used in the six field trails. The varieties are grouped according to the Italian bread wheat classification, which is based on specific rheologic and analytical parameters [31]. FF: improver wheat; FPS: superior bread making wheat; FP: ordinary bread making wheat; FB: wheat for biscuits.

Italian Bread Wheat Classification	Genotype Name	Field Trials
FF	Aiace_VST	2019–2020
	Bingo_VST	2019–2020
	Bologna	both years
	Giambologna	2018–2019
	Giorgione	both years
	Izalco_CS	2019–2020
	Metropolis	2018–2019
	Nestore	2018–2019
	Rebelde	both years
FPS	Albagran	2019–2020
	Berta	2019–2020
	Blasco	2018–2019
	Lancillotto	both years
	Pinturicchio	2019–2020
	RGT_Algoritmo	2018–2019
	RGT_Montecarlo	both years
	RGT_Vivendo	2019–2020
	Solindo_CS	2019–2020
	SY_Starlord	2019–2020
	SY_Ural	2019–2020
	Teorema	2018–2019
FP	Amburgo	2018–2019
	Angelico	2018–2019
	Antigua	both years
	Artù_SN	both years
	Ascott	2018–2019
	Ausonio	2018–2019
	Brunelleschi	2018–2019
	Cabiria	2019–2020
	Somtuoso_CS	both years
	Filon	2019–2020
	Ginger	2019–2020
	Guercino	both years
	KWS_Coli	2019–2020
	KWS_Lazulli	2019–2020
	LG_Absalon	2018–2019
	LG_Arnova	both years
	LG_Ascona	both years
	LG_Ayrton	2018–2019
	Poker_VST	2019–2020
	Porticcio	both years
	PR22R58	2018–2019
	Sofolk	2018–2019
	Solehio	both years
	Sothys_CS	both years
	Stromboli	both years
	SY_Capitano	2019–2020
	SY_Cicerone	both years
FB	Avila	both years
	Bramante	both years
	Canaletto	2019–2020
	Cosmic	2018–2019
	Oregrain	2018–2019
	Santorin	2018–2019

**Table 2 plants-12-01349-t002:** Geographical coordinates and sowing and harvesting dates related to all locations and agronomic seasons.

Year	Location	Field Coordinates	Sowing Date	Harvesting Date
2018/2019	Foggia	41°27′45.26” N, 15°30′11.06” E	6 December 2018	18 June 2019
2019/2020	Foggia	41°27′46.03” N, 15°30′10.02” E	9 December 2019	11 June 2020
2018/2019	Grosseto	42°57′14.332” N, 11°5′45.956” E	3 January 2019	2 July 2019
2019/2020	Grosseto	42°57′26.557” N, 11°5′47.68” E	5 January 2020	24 June 2020
2018/2019	Voghera	45°1′44.83” N, 9°1′19.189” E	25 October 2018	20 July 2019
2019/2020	Voghera	45°1′0.3” N, 8°59′44.822” E	9 December 2019	12 July 2020

## Data Availability

Not applicable.

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
