# Peer review of "Exploring Variability of Free Asparagine Content in the Grain of Bread Wheat (Triticum aestivum L.) Varieties Cultivated in Italy to Reduce Acrylamide-Forming Potential"

_plants, 2023, doi:10.3390/plants12061349_

Round 1
Reviewer 1 Report
The manuscript reports about the accumulation of free asparagine in the grain of different wheat genotypes, particularly 54 wheat élite varieties that are relevant for the Italian market, monitored in different environments during two sequential growing years. The fAsn concentration has been identified as the main determinant of acrylamide-forming potential in wheat and other cereals. Acrylamide, a suspected human carcinogen, is generated by Maillard cascade reactions during food processing at high temperatures, which involves reducing sugars and free asparagine.
This comprehensive analysis is quite interesting considering the possibility of identifying the specific bread wheat varieties with characteristic of good stability among a range of environments and harvest years in fAsn concentration, while maintaining the agronomic and bread making quality.
In regard to scientific content, a few aspects should be reconsidered.
To my opinion, box plots of fAsn measurements in the six considered field trials and from the 18 genotypes present in the six field trials (Figure 3 and 4) are lacking in numeric scale for fAsn concentration (mmol/Kg) on y axis.
As far as the writing of the text:
Line 177: 2.2. Evaluation of a the influence of growing environments on fAsn accumulation should be checked.
I suggest using uniform style for content of fAsn throughout the paper: mmol/kg vs mmol/Kg
Author Response
The manuscript reports about the accumulation of free asparagine in the grain of different wheat genotypes, particularly 54 wheat élite varieties that are relevant for the Italian market, monitored in different environments during two sequential growing years. The fAsn concentration has been identified as the main determinant of acrylamide-forming potential in wheat and other cereals. Acrylamide, a suspected human carcinogen, is generated by Maillard cascade reactions during food processing at high temperatures, which involves reducing sugars and free asparagine.
This comprehensive analysis is quite interesting considering the possibility of identifying the specific bread wheat varieties with characteristic of good stability among a range of environments and harvest years in fAsn concentration, while maintaining the agronomic and bread making quality.
We thank the reviewer for having appreciated our study and for the pertinent observations that are helping us to improve the manuscript. We are going to explain, point by point, the details of the revisions to the manuscript.
1 - In regard to scientific content, a few aspects should be reconsidered.
To my opinion, box plots of fAsn measurements in the six considered field trials and from the 18 genotypes present in the six field trials (Figure 3 and 4) are lacking in numeric scale for fAsn concentration (mmol/Kg) on y axis.
As requested, we added the numeric scale for fAsn concentration (mmol/Kg) in Figures 3 and 4.
2 - As far as the writing of the text:
Line 177: 2.2. Evaluation of a the influence of growing environments on fAsn accumulation should be checked.
We corrected the paragraph title
3 - I suggest using uniform style for content of fAsn throughout the paper: mmol/kg vs mmol/Kg
We uniformed the style and used mmol/kg throughout the text.
We also corrected some minor typing errors throughout the text.
All the revisions are marked up using the “Track Changes” function of MS Word.
Reviewer 2 Report
English translation:
Acrylamide has been a hot topic in recent years among health food promoters,food manufacturers and consumers. The research is extensive, many varieties
of wheat harvested from the same geographical area were analyzed. The variation
of environmental factors on grain quality is important, and the researchers
took this into account. It was a pleasure to read this study and I agree with the publication.
Original:
Acrilamida a fost un subiect fierbinte în ultimii ani în rândul promotorilor de alimente sănătoase, producătorilor de alimente È™i consumatorilor. Cercetarea este amplă, au fost analizate multe soiuri de grâu recoltate din aceeaÈ™i zonă geografică. VariaÈ›ia factorilor de mediu asupra calității cerealelor este importantă, iar cercetătorii au È›inut cont de acest aspect.
A fost o plăcere să citesc acest studiu și sunt de acord cu publicația.
Author Response
We thank the reviewer for having appreciated our study. Minor revisions have been made according to suggestions from Reviewer 1.